First microsatellite markers for the pine catkin sawfly Xyela concava (Hymenoptera, Xyelidae) and their application in phylogeography and population genetics

Kulanek Dustin dustin.kulanek@senckenberg.de
Blank Stephan M.
Kramp Katja
Senckenberg Deutsches Entomologisches Institut , Müncheberg , Germany
Uversky Vladimir
Electronic publication date: 2019 Nov 13
Publication date: 2019
Volume: 7
Electronic Location ID: e8010
Received 2019 May 16; Accepted 2019 Oct 8
Copyright: ©2019 Kulanek et al.
Copyright year: 2019
Copyright holder: Kulanek et al.
License: This is an open access article distributed under the terms of the Creative Commons Attribution License, which permits unrestricted use, distribution, reproduction and adaptation in any medium and for any purpose provided that it is properly attributed. For attribution, the original author(s), title, publication source (PeerJ) and either DOI or URL of the article must be cited.
License URL: https://creativecommons.org/licenses/by/4.0/

Keywords: Xyela concava, Microsatellites, Population genetics, Phylogeography, Sawflies, Xyelidae, Mitochondrial introgression

Funding: The authors received no funding for this work.

==============================
Microsatellites are widely used as powerful markers in population genetics because of their ability to access recent genetic variation and to resolve subtle population genetic structures. However, their development, especially for non-model organisms with no available genome-wide sequence data has been difficult and time-consuming. Here, a commercial high-throughput sequencing approach (HTS) was used for the very first identification of microsatellite motifs in the genome of Xyela concava and the design of primer pairs flanking those motifs. Sixteen of those primer pairs were selected and implemented successfully to answer questions on the phylogeography and population genetics of X. concava. The markers were characterized in three geographically distinct populations of X. concava and tested for cross-species amplification in two additional Xyela and one Pleroneura species (Xyelidae). All markers showed substantial polymorphism as well as revealing subtle genetic structures among the three genotyped populations. We also analyzed a fragment of the nuclear gene region of sodium/potassium-transporting ATPase subunit alpha (NaK) and a partial mitochondrial gene region coding for cytochrome oxidase subunit I (COI) to demonstrate different genetic resolutions and sex-biased patterns of these markers, and their potential for combined use in future studies on the phylogeography and population genetics of X. concava. Although a limited number of populations was analyzed, we nevertheless obtained new insights on the latter two topics. The microsatellites revealed a generally high gene flow between the populations, but also suggested a deep historical segregation into two genetic lineages. This deep genetic segregation was confirmed by NaK. While the high gene flow was unexpected, because of assumed restricted dispersal ability of X. concava and the discontinuous distribution of the host trees between the populations, the segregation of two lineages is comprehensible and could be explained by different refuge areas of the hosts during glacial times. The COI results showed a discordant strong genetic structure between all populations, which might be explained by the smaller effective population size of the mitochondrial genome. However, given the frequent evidence of a similar nature in recent studies on sawflies, we also consider and discuss mitochondrial introgression on population level as an alternative explanation.

Introduction

Xyelidae have always attracted the attention of taxonomists and systematists. They represent the sister group of the rest of the megadiverse insect order Hymenoptera (Ronquist et al., 2012; Klopfstein et al., 2013; Malm & Nyman, 2015), which is traditionally divided into the paraphyletic “Symphyta” (missing the wasp waist) and the monophyletic Apocrita (sharing a wasp waist as a derived feature ) (Malm & Nyman, 2015). The recent inconsistent phylogenetic placement of xyelids together with Pamphiliidae and Tenthredinidae as sister clade to all remaining Hymenoptera by Peters et al. (2017) might have been caused by an artificial grouping due to shared very slow mutation rates in those groups (Ronquist et al., 2012).

The rich fossil record of Xyelidae includes the earliest fossil forms of Hymenoptera dating from the Middle–Upper Triassic (Kopylov, 2014). Proper knowledge of the phylogeography and population genetics of xyelids is therefore important in understanding the underlying evolutionary processes, which in turn will help to understand the evolution of other hymenopterans. Unfortunately, such data are scarce for xyelids due to the rarity of many species, ephemerality of the imagines, and considerable problems in identifying species morphologically as well as genetically (e.g., Burdick, 1961; Blank, Shinohara & Byun, 2005; Blank, Shinohara & Altenhofer, 2013; Blank & Kramp, 2017; Blank, Kramp & Shinohara, 2017; Blank et al., 2017). While a limited number of microsatellite studies has been conducted on sawflies (Hartel, Frederick & Shanower, 2003; Cook et al., 2011; Caron et al., 2013; Bittner et al., 2017), none has focused on xyelids. Consequently, little is known about the population dynamics of this early-diverging group, including effects of ephemerality of imagines and their dispersal ability, host adaption and host dependency, and reproduction mode.

Here, we report on the first developed and tested set of 16 polymorphic nuclear microsatellite markers for Xyela concava (Burdick, 1961), to shed light on the latter issues. X. concava is widely distributed in southwestern USA, where it is closely associated with the pinyon-juniper woodland vegetation type of higher elevation semideserts, i.e., with pine species of the subgenus Strobus subsection Cembroides (Farjon, 2010). Females oviposit into developing male cones of Pinus cembroides Zuccarini, 1830, P. edulis Engelmann, 1848 and P. monophylla Torrey & Frémont, 1845 (Fig. 1), where the larvae feed on the sporophylls. After having ceased feeding, Xyela larvae dig into the soil below the host trees and may diapause up to several years before pupating (Blank, Shinohara & Altenhofer, 2013). Imagines of the next generation emerge during spring and often visit flowering plants with easily accessible anthers, such as mountain mahogany (Cercocarpus spp.) and cliff-rose (Purshia spp.), from which they gather pollen for nutrition with their adapted mouthparts (Burdick, 1961; SM Blank, pers. obs., 2013–2017). Flight behavior is described as erratic and slow (Burdick, 1961; Aron et al., 2005). Therefore, given an assumed restricted dispersal ability and a close association with particular host species, it is intriguing to see how the variation within and among populations have been influenced by the distribution of the host trees during glacial and postglacial times. The high resolution power and therefore high capability of microsatellite markers to assess subtle and recent population genetic structures makes them well suited to this task. We used a commercial high-throughput sequencing approach for the development of the microsatellite markers and applied them to describe genetic structures and variation among and within three geographically distinct populations of X. concava. Furthermore, we compared the resolution of genetic variation of these markers with compiled data for one nuclear and one mitochondrial gene coding region and discuss their possible combined suitability for identifying genealogical lineages and answering phylogeographical questions. Finally, cross-amplification patterns for two species of Xyela and one of Pleroneura, sister taxon of Xyela (Smith, 1967), are illustrated.

Figure 1 Location of the collection areas and distribution of the host species.

Credit Pinus spec. shape files: https://data.usgs.gov/metadata.

Material and Methods

Sampling

Xyela larvae were extracted from staminate cones of pines as described by Blank, Shinohara & Altenhofer (2013) and stored in 100% ethanol at −20 °C. We included in the analysis larvae originating from three collection sites which are located 900–1,200 km from each other (see Table S1). The specimens are preserved in the Senckenberg Deutsches Entomologisches Institut, Müncheberg, Germany. Since it is impossible to identify Xyela larvae to species level morphologically, they were COI barcoded and identified by comparison with sequences from imagines identified as X. concava morphologically (identification following Burdick (1961), reference sequences of imagines were published by Blank, Kramp & Shinohara (2017) and are deposited in the GenBank (NCBI) database, accession numbers KY198313 and KY198314). Finally, 98 larvae of X. concava were selected for the analysis (for detailed data see Table S1).

DNA extraction

Whole larvae were used for DNA extraction. The integument was slightly cut with a scalpel, so that the exterior stayed intact for later morphological inspection. DNA was extracted and purified with E.Z.N.A. Tissue DNA Kit (Omega Bio-Tek) according to the manufacturer’s protocol, but with an extended 2 h incubation time at 55 °C (Thermomixer, without shaking) for cell lysis. The extracted DNA was stored at −20  °C until later use. The integuments were retained and stored in 70% ethanol.

Microsatellite marker development and screening

Total genomic DNA of a single female of X. concava (specimen ID: DEI-GISHym 30887, see Table S1) was extracted following the protocol described above. 10 ng/µl DNA in a total volume of 20 µl was sent to AllGenetics & Biology (Coruña, Spain) for the commercial identification of microsatellite motifs and the design of motif flanking primer pairs. A library was prepared for the DNA sample using the Nextera XTDNA kit (Illumina), following the manufacturer’s instructions. The library was enriched with the following microsatellite motifs: AC, AG, ACG, and ATCT. Enriched DNA was sequenced in the Illumina MiSeq platform (PE300) and produced 3,043,190 paired-end reads. These paired-end reads were processed in Geneious 10.0.5 (Biomatters, Ltd.) using in-house developed scripts (property of AllGenetics & Biology) and overlapped into 1,521,595 sequences (trim error probability limit of 0.03). Primer design was carried out by AllGenetics & Biology in Primer 3 (Koressaar & Remm, 2007; Untergasser et al., 2012) for 500 sequences containing microsatellite motifs. For a preliminary screening, fifty primer pairs were picked and four X. concava larvae (DEI-GISHym 32824–32827) were used for tests of polymorphism. Furthermore, 12 specimens of X. deserti (Burdick, 1961), 12 specimens of an undescribed Xyela species, possibly a member of the X. alpigena group (Blank & Kramp, 2017), and six specimens of Pleroneura koebelei Rohwer, 1910 (see Table S1 ) were tested for cross-species amplification to check the marker system for potential use on two closely and one more distantly related xyelid species. The PCR analysis included a temperature gradient in the primer annealing step to find the best conditions for each primer pair. PCR was carried out in a total volume of 5 µl containing 0.5 µl DNA, 0.1 µl of primers (10 pmol each) and 2.5 µl of 2×Multiplex PCR Plus Master mix (QIAGEN). The PCR protocol consisted of an initial DNA polymerase (HotStar Taq) activation step at 95 °C for 5 min, followed by 35 cycles of 30 s of 95 °C (DNA denaturation step), 90 s at 50 °C, 52 °C, 54 °C and 56 °C (primer annealing step, temperature ramp), and 30 s at 72 °C (elongation step); the last cycle was followed by a final 10 min extension step at 68 °C. 5 µl of the PCR product was visualized on a 2 % agarose gel. Primer pairs that produced no amplification, multiple or unexpected size PCR products were discarded. Eighteen primer pairs, showing discernably strong and specific signals, were picked for further analysis. 5′-end fluorescently labelled reverse primers (6-Fam (Biomers) and NED, VIC, PET (Thermo Fisher Scientific)) for the selected primer pairs were synthesized for multiplexing and capillary electrophoresis. PCR was carried out in four multiplex reactions for four X. concava DNA samples in a total volume of 10 µl containing 2.5 µl DNA, 1.0 µl of fluorescently labelled primer pair mix (0.5 pmol each, containing up to five primer pairs, depending on compatible annealing temperature, dye and expected fragment size range) and 5.0 µl of 2×Multiplex PCR Plus Master mix (QIAGEN). PCR reaction conditions were as described above with the respective optimal annealing temperature for each primer pair mix. Reactions were diluted 1:2 and sent to Macrogen Europe (Amsterdam, the Netherlands) for fragment analysis.

Allele sizes were scored using GeneMapper 5.0 (Applied Biosystems). No marker showed strong stutter peaks or intensive background signal. Two primer pairs appeared to be monomorphic and were excluded from further analyses. Sixteen primer pairs showed apparent polymorphism for the four tested samples and were finally selected (Table 1).

COI and NaK polymerase chain reaction analysis

Primers used for amplification and sequencing are listed in Table 2. The mitochondrial region amplified is a 1,078 bp long fragment of cytochrome oxidase subunit I gene (COI). The first 658 bp of this fragment (from the 5′ end) correspond to the standard barcode region of the animal kingdom (Hebert et al., 2004). Additionally, a 1,654 bp long fragment of the nuclear gene region of sodium/potassium-transporting ATPase subunit alpha (NaK) was amplified.

PCR reactions were carried out in a total volume of 20–25 µl containing 1.5–3.0 µl DNA, 1.2–2.5 µl of primers (5 pmol each) and 10.0–12.5 µl of 2×Multiplex PCR Plus Master mix (QIAGEN). The PCR protocol consisted of an initial DNA polymerase (HotStar Taq) activation step at 95 °C for 5 min, followed by 38–40 cycles of 30 s at 95 °C, 90 s at 49–59 °C depending on the primer set used, and 50–120 s (depending on the amplicon size) at 72  °C; the last cycle was followed by a final 30 min extension step at 68 °C. 3 µl of the PCR product was visualized on a 1.4 % agarose gel. Primers and dNTPs were inactivated with FastAP and Exonuclease I (Thermo Fisher Scientific). 1.7–2.2 U of both enzymes were added to 17–22 µl of PCR solution and incubated for 15 min at 37 °C, followed by 15 min at 85 °C. Purified PCR products were sent to Macrogen Europe (Amsterdam, the Netherlands) for sequencing. To obtain unequivocal sequences, both sense and antisense strands were sequenced. Sequences were aligned manually with Geneious 11.0.5. Ambiguous positions (i.e., double peaks in chromatograms of both strands) due to heterozygosity or heteroplasmy were coded using IUPAC symbols. Sequences have been deposited in the GenBank (NCBI) database (accession numbers MK265017 –MK265114 and MK264919 –MK265016, for detailed data see Table S1).

Table 1 Sixteen polymorphic microsatellite loci and the corresponding flanking primer pairs identified in the pine catkin sawfly Xyela concava.

Locus	Size range (bp)	Motif	Ta in °C	Label	Primer sequence (5′—3′)	
AG_30887_445	75–93	AAG(11)	50	VIC	F: GTCTCGACTCCCTCCTACGA	
					R: ACGGAAGTGCATCGGATCTTC	
AG_30887_046	195–225	AGC(30)	50	PET	F: CCTTTCGTCCTGGTTGACCA	
					R: GATACGCCAGCCTATCCGTC	
AG_30887_083	178–190	AAG(10)	50	6-Fam	F: TTCCAGTTTCTTGCAACGCG	
					R: ATTCGCAAGCCTCTTCTGCA	
AG_30887_188	179–188	AAT(9)	50	NED	F: GCGGCGGTATAATGAGTCGT	
					R: GGAAAGTGACTGCTACCGGT	
AG_30887_479	93–102	ACT(8)	50	PET	F: GCTGTTCACATGGCAGGTAG	
					R: CCACCATCCCTACTACGGCT	
AG_30887_193	110–134	AGC(17)	50	VIC	F: AGAGTGCCAACGTGGGAAAT	
					R: TTACTTTGCCCATGCCATGC	
AG_30887_234	376–424	AATGCG(8)	50	PET	F: AGTCTGATCCTTCCTGCGGA	
					R: ATACGTGCCAGTTCGATCGT	
AG_30887_282	239–263	AGC(10)	50	6-Fam	F: CTGTGCCTACGTCCCTTAGG	
					R: CCCATCGTTTGGTCGGTAGA	
AG_30887_286	103–121	AGC(8)	50	NED	F: GCGTCCGTCTGAAATCTTGG	
					R: CATTCGCATTCGACGCACTC	
AG_30887_179	111–126	AGC(9)	50	6-Fam	F: CCCGTTCGTAAATCGGTCCT	
					R: GACGTGGAATCGGTGGACTC	
AG_30887_460	90–116	AT(5)	50	PET	F: ACGTACTTATTGGGCGCGAA	
					R: TTTACATGCTGTACACCGGGA	
AG_30887_347	237–249	AAG(8)	50	PET	F: CCCGGACCTCGTGCTATTC	
					R: GGCGACAATCCCACGTGATA	
AG_30887_393	136–175	AAG(8)	50	6-Fam	F: CCATCACTGTGCCGCGATAT	
					R: GCACCTCAGGGATCCTCAAT	
AG_30887_414	122–179	AAG(8)	50	NED	F: TGATTTGTGCAACCGAGGGA	
					R: CCCTTTATTCTCAGCAACCGC	
AG_30887_012	130–148	AGG(9)	50	PET	F: TTCCGGACGACTTTGACCTG	
					R: CCTCGATTCCGATTCCCGTT	
AG_30887_223	120–186	AAG(9)	50	6-Fam	F: TCAAAGCGGAGAAAGAGCGT	
					R: TTAACCGCCATCGACCGTTC	

Table 2 Nuclear NaK and mitochondrial COI primers used for amplification (PCR) and sequencing (seq).

Gene region	Primer name	Primer sequence (5′-3′)	Ta in °C	PCR/ Sequencing	Reference	
COI	symF1	TTTCAACWAATCATAAARAYATTGG	49	PCR, seq	Prous et al. (2016)	
COI	symR1	TAAACTTCWGGRTGICCAAARAATC	49	PCR/ seq	Prous et al. (2016)	
COI	symC1-J1751	GGAGCNCCTGATATAGCWTTYCC	49	seq	Prous et al. (2016)	
NaK	NaK_263F	CTYAGCCAYGCRAARGCRAARGA	59	PCR/ seq	Prous et al. (2017)	
NaK	NaK_907Ri	TGRATRAARTGRTGRATYTCYTTIGC	59	seq	Prous et al. (2017)	
NaK	NaK_1250Fi	ATGTGGTTYGAYAAYCARATYATIGA	59	seq	Prous et al. (2017)	
NaK	NaK_1918R	GATTTGGCAATNGCTTTGGCAGTDAT	59	PCR/ seq	Prous et al. (2017)	

Genetic data analysis

Estimations of genetic variation were obtained by calculating average number of alleles (NA), observed (HO) and expected heterozygosity (HE) as well as deviations from Hardy-Weinberg equilibrium (HWE) for each locus for all X. concava populations using ARLEQUIN 3.5.2.2 (Excoffier & Lischer, 2010) and 1,000 permutations. The same program was used to assess the suitability of resolving population differentiation by estimating population pairwise measures of FST (1,000 permutations). The program GENEPOP 4.7.0 (Rousset, 2008) was used to estimate the inbreeding coefficient FIS (1,000 permutations). GENEPOP was also used in combination with the ENA correction implemented in the program FreeNA (Chapuis & Estoup, 2007) to test for the presence and frequency of null alleles in the populations and to correct for the potential overestimation of FST values induced by the occurrence of null alleles (1,000 permutations). Number of genotypes (NG) in the dataset was counted with Excel. To test for isolation by distance, a Mantel test for the microsatellite data was performed (1,000 replicates) in ALLELES IN SPACE (Miller, 2005).

To assess the suitability of the microsatellite markers for assessing genetic population structures, three independent Bayesian assignment tests were carried out, one non-spatial using STRUCTURE 2.3.4 (Pritchard, Stephens & Donnelly, 2000) and two spatial model based using BAPS 6.0 (Corander, Waldmann & Sillanpää, 2003; Corander, Sirén & Arjas, 2008) and GENELAND 4.0.8 (Guillot, Mortier & Estoup, 2005). GENELAND assignment results for the microsatellite markers were also compared with results in GENELAND for the mitochondrial and nuclear gene coding markers (here without any comparison with a non-spatial assignment in STRUCTURE, since the model assumptions are likely to be violated for sequence data (Falush, Stephens & Pritchard, 2003). In BAPS, a maximum number of K = 10K was given as a prior. In STRUCTURE, ten replicates for each K from 1 to 10 were carried out with 50,000 burn-in steps followed by 100,000 MCMC. The online program STRUCTURE HARVESTER (Earl & vonHoldt, 2012) was used to infer the most likely value of K. GENELAND was carried out with an uncertainty on coordinates of 25 km, 100,000 iterations, a thinning to every 100 replicate and 10 independent runs. In STRUCTURE and GENELAND, a no admixture model and independency of allele frequency (uncorrelated model) was assumed, since correlated frequency models, though more powerful in detecting subtle differentiations, are more sensitive to departure from model assumptions (Guillot et al., 2012).

Results

The identification of microsatellite motifs by using HTS yielded 500 potential markers of which 50 were picked for a preliminary screening. Sixteen were finally implemented. Alongside primer pairs that produced no amplification or were monomorphic, some also unexpectedly showed PCR products of multiple sizes and had to be discarded.

The microsatellite markers amplified 3–14 different alleles and 3–18 different genotypes per population and locus (Table 3). Observed heterozygosities ranged from 0.00 to 0.78 and were significantly lower than those expected under Hardy-Weinberg equilibrium except for one locus, indicating a deficiency of heterozygotes in the analyzed Xyela concava populations and/or the presence of null alleles. This deficit is also confirmed by positive FIS values obtained for all but three loci in one population. Estimated frequencies of null alleles were variable depending on the respective microsatellite locus and X. concava population and varied between 0 and 39 % (Table 4).

Table 3 Comparative genetic diversity values for the three Xyela concava populations.

Analyzed for each of the 16 microsatellite loci and on average over all loci including number of alleles (NA), Number of genotypes (NG), observed (HO) and expected (HE) heterozygosity and estimates of FIS.

Locus	Big Burro Mountains	Monitor Pass	Uinta Mountains	
	NA/NG	FIS	HO	HE	NA/NG	FIS	HO	HE	NA/NG	FIS	HO	HE	
AG_30887_445	6/7	0.91	0.07	0.78*	6/11	0.50	0.40	0.79*	7/12	0.43	0.40	0.69*	
AG_30887_046	10/11	0.33	0.61	0.85*	9/14	0.44	0.47	0.82*	7/11	0.04	0.78	0.80*	
AG_30887_083	5/7	0.39	0.43	0.69*	3/5	0.63	0.20	0.53*	4/5	−0.18	0.63	0.53*	
AG_30887_188	4/6	0.84	0.11	0.66*	3/5	0.31	0.37	0.53*	3/5	0.63	0.25	0.66*	
AG_30887_479	3/4	0.22	0.43	0.54*	4/5	0.79	0.10	0.47*	4/6	−0.06	0.53	0.49*	
AG_30887_193	6/8	0.31	0.50	0.72*	5/12	0.47	0.40	0.74*	7/13	0.04	0.78	0.80*	
AG_30887_234	6/9	0.34	0.50	0.75*	6/8	0.42	0.40	0.68*	6/9	0.21	0.63	0.78*	
AG_30887_282	8/8	0.40	0.46	0.77*	6/9	0.61	0.30	0.75*	6/9	−0.03	0.73	0.70*	
AG_30887_286	6/8	0.76	0.18	0.74*	5/9	0.24	0.47	0.61	7/11	0.53	0.35	0.74*	
AG_30887_179	3/3	1.00	0.00	0.62*	5/7	0.55	0.20	0.43*	5/6	0.76	0.15	0.61*	
AG_30887_460	6/6	0.75	0.14	0.55*	4/4	0.30	0.13	0.18*	6/6	0.74	0.15	0.56*	
AG_30887_347	4/5	0.34	0.43	0.64*	3/6	0.51	0.33	0.67*	4/7	0.06	0.63	0.66*	
AG_30887_393	7/7	0.82	0.11	0.59*	6/10	0.44	0.40	0.71*	5/9	0.67	0.20	0.59*	
AG_30887_414	12/12	0.35	0.54	0.82*	10/18	0.54	0.40	0.86*	9/13	0.13	0.68	0.77*	
AG_30887_012	5/7	0.90	0.07	0.73*	3/4	0.51	0.27	0.54*	3/4	0.67	0.23	0.67*	
AG_30887_223	9/11	0.76	0.14	0.80*	14/18	0.36	0.47	0.89*	13/15	0.72	0.23	0.82*	
Mean		0.59	0.29	0.71		0.48	0.33	0.64		0.33	0.46	0.68	
S.D.		0.26	0.21	0.09		0.13	0.12	0.18		0.33	0.24	0.10	
Notes.

* significant departure from H-W equilibrium (P < 0.05).

S.D., standard deviation.

Table 4 Estimated null allele frequencies for each of the 16 polymorphic microsatellite loci and each population including the average null allele frequency.

Estimated null allele frequency	
Locus	Big Burro Mts	Monitor Pass	Uinta Mts	
AG_30887_445	0.395	0.221	0.167	
AG_30887_046	0.165	0.191	0.028	
AG_30887_083	0.175	0.229	0.041	
AG_30887_188	0.334	0.116	0.247	
AG_30887_479	0.095	0.267	0.040	
AG_30887_193	0.130	0.194	0.037	
AG_30887_234	0.161	0.184	0.087	
AG_30887_282	0.194	0.260	0.036	
AG_30887_286	0.314	0.073	0.208	
AG_30887_179	0.381	0.190	0.282	
AG_30887_460	0.259	0.000	0.257	
AG_30887_347	0.148	0.200	0.048	
AG_30887_393	0.309	0.163	0.247	
AG_30887_414	0.196	0.245	0.053	
AG_30887_012	0.378	0.183	0.264	
AG_30887_223	0.319	0.162	0.314	
Mean	0.247	0.180	0.147	

The estimation of the frequency of null alleles, though highly variable depending on the locus-population combination, did not introduce any bias to our dataset and thus did not cause an overestimation of pairwise FST values.

The FST values uncorrected and corrected for the presence of null alleles showed higher values between the populations of Monitor Pass and Uinta Mountains as well as between the populations of Monitor Pass and Big Burro Mountains than the values between the populations of Uinta Mountains and Big Burro Mountains (Table 5). In general, all FST values were comparatively low (0.028–0.113) but either had a considerably narrow confidence interval or were significant or approaching the level of significance (P = 0.055). The FST values for NaK and COI were, in comparison, higher (0.215–0.740). While the values for NaK showed the same pattern as the microsatellite markers in respect of genetic relationship of the populations, the FST values for COI indicated relatively high differences between all populations (Table 6). The Mantel test showed no isolation by distance (r2 = 0.0173, P <0.001). While spatial assignment tests for NaK and the mircosatellites came up with the same pattern as the FST values—as indicated by the assignment of two populations with high posterior probabilities to one genetic group or lineage (Figs. 2 and 3)—the non-spatial STRUCTURE analysis for microsatellites was slightly non-confirmative, with genotypes from the Big Burro Mountains and Uinta Mountains assigned to one separate lineage, yet with genotypes from all three populations assigned to one shared overlapping genetic lineage. The analysis of the COI data revealed that each population represented one distinct cluster (K = 3) (Figs. 3B–3D).

Table 5 Pairwise FST estimates between populations of Xyela concava for the 16 microsatellite loci including corresponding P values and confidence intervals.

Estimates are given both uncorrected and corrected for the presence of null alleles. Bold typeface denotes pairwise FST estimates that are significantly different from zero (P < 0.005). Values in square brackets indicate 95% confidence intervals for pairwise corrected FST estimates.

FST uncorrected	Big Burro Mts	Monitor Pass	Uinta Mts	
Big Burro Mts	*			
Monitor Pass	0.09182	*		
Uinta Mts	0.02254	0.07705	*	
FSTENA corrected	Big Burro Mts	Monitor Pass	Uinta Mts	
Big Burro Mts	*			
Monitor Pass	0.083 [0.054, 0.115]	*		
Uinta Mts	0.015 [0.004, 0.028]	0.065 [0.041, 0.094]	*	

Table 6 Pairwise FST estimates between populations of Xyela concava for NaK and COI including corresponding P values.

Bold typeface denotes pairwise FST estimates that are significantly different from zero (P < 0.005).

	Big Burro Mts	Monitor Mts	Uinta Mts	
NaK				
Big Burro Mts	*			
Monitor Pass	0.740	*		
Uinta Mts	0.215	0.680	*	
COI				
Big Burro Mts	*			
Monitor Pass	0.699	*		
Uinta Mts	0.508	0.678	*	

Figure 2 Bayesian assignment of Xyela concava populations to each of the identified clusters (K= 2) for the microsatellite markers.

(A) GENELAND (Posterior probabilities are indicated in the scale bar. The contour lines in the maps indicate the spatial positions of genetic discontinuities. Lighter shading indicates a higher probability of belonging to the genetic population), (B) BAPS (the area of each population is proportional to the number of specimens used) and (C) STRUCTURE.

Figure 3 Bayesian spatial assignment (GENELAND) of Xyela concava populations to each of the identified clusters for (A) NaK (K = 2) and (B), (C), (D) COI (K = 3).

The different colors represent the estimated posterior probabilities of the membership to each cluster. Posterior probabilities are indicated in the scale bar. The contour lines in the maps indicate the spatial positions of genetic discontinuities. Lighter shading indicates a higher probability of belonging to the genetic population.

All microsatellite markers were successfully tested for cross-species amplification. For the three additional species of Xyela and Pleroneura, four markers showed polymorphic products and five were apparently monomorphic for X. deserti. Eight markers showed polymorphic products for the new Xyela species of the alpigena group, while no or unspecific fragments were amplified for Pleroneura koebelei (Table 7).

Table 7 Cross-species amplification.

Locus	Xyela deserti	Xyela spec. nov.	Pleroneura koebelei	
AG_30887_445	−	−	−	
AG_30887_046	+	++	−	
AG_30887_083	−	−	−	
AG_30887_188	−	−	−	
AG_30887_479	+	++	−	
AG_30887_193	−	++	−	
AG_30887_234	+	++	−	
AG_30887_282	++	++	−	
AG_30887_286	++	++	−	
AG_30887_179	−	−	−	
AG_30887_460	−	−	−	
AG_30887_347	++	++	−	
AG_30887_393	+	−	−	
AG_30887_414	++	−	−	
AG_30887_012	−	−	−	
AG_30887_223	+	++	−	
Notes.

(−) no product

(+) monomorphic product

(+ +) polymorphic product

Discussion

Of 50 initial markers, only 16 could finally be implemented. Such high drop-out rates due to large numbers of repetitive motifs throughout the genome causing nonspecific binding of primers are already known (Schoebel et al., 2013). Other recent studies on invertebrates, using the same commercial HTS approach for the identification of SSR motifs, resulted in 11 to 21 polymorphic microsatellite markers, which nonetheless could be applied successfully (Reineke et al., 2015; González-Castellano et al., 2018; Gomes et al., 2019).

The analyses demonstrated that the degree of variability of the new microsatellite marker set is adequate in that it reveals polymorphic alleles within and across populations. The low significant deviations from Hardy-Weinberg equilibrium as well as positive FIS values for almost all loci in all populations could, however, have several causes. Given the ephemerality of the imagines and their fluctuating abundance due to extended diapausing, the major reason might have been a sampling bias, where only a fraction of each population was sampled (Wahlund effect). Furthermore, homozygote genotypes equally distributed across all populations indicated haploidy for altogether 26 specimens and may have had an impact on the discrepancy between the observed and expected heterozygosity. Thelytokous parthenogenesis—producing solely female offspring—which is known in xyelids (Blank, Shinohara & Altenhofer, 2013), also might have contributed to the deficiency. However, due to the observed genotypic variation across the data set, apomictic parthenogenesis seems unlikely (Caron et al., 2013).

The results based on the non-spatial model in STRUCTURE were not as confirmative as in the spatial-model based assignment tests. Since in STRUCTURE no spatial information and therefore fewer assumptions are incorporated, geographical barriers and distance as likely causes for differentiated populations might have been underestimated (Coulon et al., 2006). On the other hand, because it does not include spatial information, STRUCTURE may here indicate a subtler genetic structure with possible higher exchange rates of the nuclear genome among all populations. However, both model applications told a broadly concordant narrative for the microsatellite markers, which are also supported by the low but significant FST values. First, the recent, seemingly discontinuous distribution of the hosts, Pinus edulis and P. monophylla, at higher elevations in mountain ranges with up to 100 km between single patches, apparently does not represent a barrier for recent and present gene flow. This is also supported by the Mantel test, which indicated no isolation by distance. X. concava is assumed to be relatively stationary due to the observed slow and erratic flight behavior (Burdick, 1961). Therefore, other explanations for the ability to disperse over long distances should be considered, such as passive dispersal by wind.

Second, the proposed geographically remote and restricted refugia of the host species during glacial times (Bentancourt et al., 1991; Grayson, 2011; Duran, Pardo & Mitton, 2012), and the considerably long distances between them, might have been sufficiently great to cause restricted gene flow and genetic segregation into two lineages. This assumption was also supported by the high and significant FST values and the genetic clustering of the NaK coding region, which due to the slower mutation rates presumably better reflects events in the past. In the FST statistics and assignment tests of the microsatellite data (displaying presumably more recent events) the segregation could still be detected, but also a recent state of admixture was indicated. To test this hypothesis and a possibly ongoing admixture of the segregated lineages, populations of X. concava in hybrid zones and overlapping distribution areas of the host species should be included in future studies.

Compared to the results of the nuclear microsatellites and NaK, FST values and Bayesian statistics for the mitochondrial COI region showed a clear non-congruent pattern with a strong genetic structure among all three populations. One explanation could be the different effective population sizes (Ne) of the mitochondrial and nuclear genome, which will cause differences in the diversity of the genetic structure of populations over time (incomplete lineage sorting; Harrison, 1989; Funk & Omland, 2003). However, this non-congruent pattern also might have been caused by biased mitochondrial introgression as often found in “Symphyta” as recently discussed by Prous, Lee & Mutanen (2019, preprint). The authors assume that mitochondrial introgression in sawflies might be promoted by a combination of the haplodiploid reproduction system of Hymenoptera (Aron et al., 2005) and the low mitochondrial mutation rates in sawflies (Tang et al., 2019). The assumption is partly based on theoretical models of Patten, Carioscia & Linnen (2015) showing that haplodiploid species are especially prone to biased mitochondrial introgression. Furthermore, Sloan, Havird & Sharbrough (2017) recently suggested that species with low mitochondrial mutation rates might favor a specific beneficial (possibly locally adapted and/or novel) mitochondrial haplotype to compensate for deleterious mitonuclear mutation loads. The specific haplotype then selectively sweeps through a population (or species) and purges deleterious mitochondrial mutations (the alternative solution being compensatory co-evolutionary changes in the nuclear genome). Tendentially, this would lead to a strong mitochondrial population structure and a mitonuclear discordance, which might be reflected in the data set. Given the evidence for the very low evolutionary rates of molecular characters in xyelids (Ronquist et al., 2012), this might be especially true for them. Additionally, mitochondrial introgression might likely be the cause for mitonuclear discordance in cases where there is a general agreement among large numbers of nuclear loci but discordance with mitochondrial genealogies (Sloan, Havird & Sharbrough, 2017). Therefore, this new set of microsatellites may also be an attractive tool to indicate mitochondrial introgression at the population level of X. concava and other closely related xyelids.

Conclusions

The implemented new set of microsatellite markers will be valuable for future analyses of additional and less distantly located populations while unraveling the population structure of Xyela concava. Together with other nuclear gene coding markers it can be used to elucidate both old and recent divisions in the gene pool to reveal more details of the phylogeography of this species. Furthermore, especially because of different underlying evolutionary processes affecting the nuclear and mitochondrial genome, this new set of microsatellites can potentially be used to reveal processes such as mitochondrial introgression at population level.

Even from this small data set, some tentative phylogeographic trends can be stated for X. concava. This study covers only three populations but nevertheless indicates a segregation of two genetic lineages and a recent state of admixture, which might have been caused by glacial retreat events. This would agree with proposed geographically separate glacial refugia of the host species. However, more populations covering the complete distribution of X. concava, especially populations from overlapping distribution areas of the hosts, need to be analyzed to test this hypothesis.

Supplemental Information

Table S1 Detailed data for Specimens of Xyela concava used for the development and for screening of the microsatellites

Total number of individuals per collection site (N), unique specimen identifier (DEI-GISHym), GenBank accession number, and collection data of the analyzed imagines (♀♂) and larvae (L) of Xyela concava

Click here for additional data file.

Table S2 Detailed data for allele sizes for all analyzed microsatellite loci and populations of Xyela concava

Unique specimen identifier (DEI-GISHym). Allele sizes in base pairs (bp). Empty cells indicate no amplification

Click here for additional data file.

We are grateful to C. Kutzscher (SDEI Müncheberg) for joining S.M. Blank during field work and for his support in the genetic lab. We thank A. Liston (SDEI Müncheberg) for a linguistic check of an earlier draft of the manuscript. We acknowledge the improvement of the manuscript by S.K. Monckton (Toronto) and an anonymous referee.

Additional Information and Declarations

Competing Interests

Author Contributions

DNA Deposition

Data Availability

The authors declare there are no competing interests.

Dustin Kulanek conceived and designed the experiments, performed the experiments, analyzed the data, contributed reagents/materials/analysis tools, prepared figures and/or tables, authored or reviewed drafts of the paper, approved the final draft.

Stephan M. Blank conceived and designed the experiments, contributed reagents/materials/analysis tools, authored or reviewed drafts of the paper, approved the final draft.

Katja Kramp conceived and designed the experiments, performed the experiments, analyzed the data, contributed reagents/materials/analysis tools, authored or reviewed drafts of the paper, approved the final draft.

The following information was supplied regarding the deposition of DNA sequences:

The COI and NaK sequences are available at GenBank: MK264919 to MK265114.

The following information was supplied regarding data availability:

The raw data are available in the Supplemental Files and at GenBank.

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
