# Peer review of "First microsatellite markers for the pine catkin sawfly Xyela concava (Hymenoptera, Xyelidae) and their application in phylogeography and population genetics"

_PeerJ, doi:10.7717/peerj.8010_

## Round 0.1 · original submission · Major Revisions

Please address all the critiques of both reviewers and revise the manuscript accordingly.

·

Basic reporting

No comment.

Experimental design

I felt that the motivation for the study was elegant, straightforward, and clear. I do think that the authors could more directly identify the knowledge gap that is being filled by the study (i.e. first microsatellite set for Xyelidae, as commented on line 43).

I also greatly appreciate the thoroughness with which the methods are explained; the provided detail is more than sufficient to enable reproduction. However, a few additional details could be provided with respect to how specimens were treated during and after DNA extraction (as commented on lines 79 & 83).

Validity of the findings

I think the results motivate a more nuanced phylogeographic discussion than what is provided; instead, the discussion seems to skirt around some possible interesting implications of the results. For instance, the authors briefly mention some phenomena which may have led to the observed population structure (lines 205-207), but provide no context as to how they might relate to the habitat, broader distribution, or population history of this species. Some of this context is given in the conclusions sections (e.g. host plant phylogeography) but I think more is needed in the discussion to help readers follow the line of thought, and to enable them to reasonably evaluate whether or not the observed subtle, recently-established population structure (lines 233-234) might be consistent with this species’ geographic context. In particular, it would be helpful to provide more information about its distribution and habitat, either in the introduction or early on in the discussion. Otherwise, to somebody unfamiliar with the study system, it’s not clear how isolated or connected the three studied populations might be. My comments on the attached PDF go into a little more detail on this matter.

I think it is also worthwhile to include some further discussion of the differences between the nuclear and mitochondrial results, particularly with regard to special considerations for the mitochondrial genomes of non-apocritans. Prous et al. (2019; preprint) briefly discuss this in the context of sawflies, for which mitochondrial introgression appears to be common. Essentially, the fact that sawflies are haplodiploid, but typically have lower mitochondrial mutation rates than those observed in Apocrita, might result in a higher rate of mitochondrial introgression between closely-related evolutionary lineages. Sloan et al. (2017) suggest that mitochondrial introgression might be one of two solutions to the problem of accumulated mitochondrial mutation load (the alternative being compensatory co-evolution of the nuclear genome); they hypothesized that mitochondrial introgression should be favoured in taxa in which mutation load accumulates slowly (e.g. sawflies). Patten et al. (2015), meanwhile, showed that hybridizing/interbreeding lineages of haplodiploid organisms are (at least mathematically) strongly biased toward mitochondrial as opposed to nuclear introgression. Taken together, one might therefore expect sawflies to exhibit artificially strong population genetic structure when measured using mitochondrial markers like COI (as mitochondrial haplotypes might tend to homogenize within a population) whereas nuclear markers might suggest less strong genetic structure between populations. Interestingly, the results presented here are consistent with these hypotheses, so I think mitochondrial introgression is at least worth a mention. The relevant references are listed below.

Patten, M. M., Carioscia, S. A., & Linnen, C. R. (2015). Biased introgression of mitochondrial and nuclear genes: A comparison of diploid and haplodiploid systems. Molecular Ecology, 24(20), 5200–5210. https://doi.org/10.1111/mec.13318

Prous, M., Lee, K. M., & Mutanen, M. (2019). Detection of cross-contamination and strong mitonuclear discordance in two species groups of sawfly genus. BioRxiv. https://doi.org/http://dx.doi.org/10.1101/525626

Sloan, D. B., Havird, J. C., & Sharbrough, J. (2017). The on-again, off-again relationship between mitochondrial genomes and species boundaries. Molecular Ecology, 26(8), 2212–2236. https://doi.org/10.1111/mec.13959

Additional comments

I made a few minor suggestions about wording in order to improve clarity. As well, minor changes to the figures are needed to improve clarity for readers with colourblindness. Other less important comments throughout the manuscript are included in the attached annotated PDF.

Overall, I very much enjoyed reading this manuscript. I find the choice of study system to be particularly interesting – which might incidentally colour my above suggestions to include more details on that topic! I apologize for submitting the review one day late; I wanted to be sure to give it sufficient time & attention. Thank you for sharing your work, and I look forward to reading the finished version.

Reviewer 2 ·

Basic reporting

no comment

Experimental design

The title is slightly misleading on the role of high throughput sequencing. In the MS 16 SSRs were developed and used in a population study. Given the approach used, 16 SSRs seems a very low return. Also, it is slightly unclear which question(s) the authors are trying to answer apart from investigation/describing the population genetic differentiation of the single species from 3 populations.

Validity of the findings

no comment

Additional comments

My main critique is that the title and the content of the MS are not well aligned with respect to "using next generation sequencing" or "new approach". The approach to use NGS to develop SSRs is really not new anymore, but can be considered standard by now. Also, the term "next generation sequencing" should not be used anymore, since a third generation is already well established. If something like that is used, then something like high throughput seq.(HTS) is more suitable. In general, the paper is mainly about using the SSRs to determine diversity and divergence in/between 3 Xyela populations, thus the title could reflect that more.
I was surprised that using HTS only yielded 16 SSRs in the end and not something in the range of 30+ at least. At 16 SSRs with a few dropout due to monomorphisms, a study could already be having too few markers.

SSR development from HTS reads (L85 etc)
This paragraphs needs substantial improvements on technical specifics, i.e. Was it single end or paired end sequencing, if the former, then specify why not the better suited paired end? How was the enrichment done?, how were SSR containing reads identified and selected? "sequenced ON the ... platform". How many reads (i.e. estimated coverage) were generated? Why were only single reads used and not more information leveraged? Specifically, I think having sequenced a diploid individual, using stacking or contig-assembly methods for the reads would provide information on which SSRs might be polymoprhic within this individual, i.e. directly yield potentially a multitude of polymorphic SSR markers. Are the reads deposited somewhere? How were they QC'ed? Which version of MiSeq sequencing kit was used? This is important because in the past 300bp sequencing runs on MiSeq had big problems for quite some time and read qualities dropped off substantially after 200bp.


Smaller remarks:
L24 specify the order "Hymenoptera" in addition to the family Xyelidae so that unfamiliar readers can relate easier.
L35 the cited paper is from 2012 and since then there have been multiple papers of more solid/comprehensive phylogenies of Hymenoptera, e.g. in Cell, which might be good to mention / check for consistency.
L86 Where is this individual from?
L97 More information on how close/far related the Pleroneura is from Xyela.
L100 Specify the gradient
L101 Are 5µl PCRs working reliably at this low volume? Which platics / PCR machine are you using for this?
L119 Here you state some markers were monomorphic. Were they not found to be polymorphic just before? Where
L132 40 cycles seems to be very high for a PCR which typically plateaus after 25-20 cycles.
L135 ExoSAP is not really a purification but more like an inactivation of relevant DNA components (dNTPs/Primers).
L206 these important aspects for Xyela biology should be also provided/discussed... in the introduction. What is known about parthenogenesis?
L211: having genotypes of individuals, which of the individuals were haploid and which were diploid, i.e. males/female? and how was ploidy considered in the analysis and how could it be influencing the results?
L217 I think the arguments against STRUCTURe should be discussed further and should include arguments from other papers looking into this. I would have assumed that maybe there is too little discriminatory power in the data and thatswhy STRUCTURE is perhaps not finding the different clusters.

Multiple places mention the names of the population according to the sampling locations. The actual names are quite meaningless for the reader, thus specify at least once, where the location is, i.e. US, Utah.

---

## Round 0.2 · Minor Revisions

Remaining critiques of the reviewer should be addressed and the manuscript should be revised accordingly.

·

Basic reporting

No comment.

Experimental design

The authors have adequately addressed my comments on the previous version of the manuscript.

Validity of the findings

The discussion is much improved and more easily followed in the revised manuscript. The authors have added sufficient context to make clear to the reader how their interpretation of the results is supported. As well, the added text about mitochondrial introgression is incorporated well, and its specific implications to this study system are very nicely highlighted.

Additional comments

I have made a few more suggestions about wording in the interest of clarity, which I leave to the discretion of the authors. Overall, it seems to me that the authors have more than adequately addressed both my comments and those of the other reviewer, and enthusiastically so. Thank you once more for sharing your work.

I have recommended minor revisions to give the authors the opportunity to address a few very minor points indicated on the attached PDF. Please use the comments panel of your PDF viewer to be sure to catch everything. I will not need to see the revised manuscript.

---

## Round 0.3 · accepted · Accept

Thank you for addressing the remaining concerns and for revising your manuscript accordingly. This version of the manuscript is acceptable now.